# Understanding the Limitations of Modifying Bitumen with Re-Refined Engine Oil Bottom (REOB)

**DOI:** 10.3390/ma18214825

**Published:** 2025-10-22

**Authors:** Lucas Mortier, Xueyan Liu, Sayeda N. Nahar, Hinrich Grothe

**Affiliations:** 1Christian Doppler Laboratory for Chemo-Mechanical Analysis of Bituminous Materials, Institute of Materials Chemistry, TU Wien, 1060 Vienna, Austria; hinrich.grothe@tuwien.ac.at; 2Section Pavement Engineering, Faculty of Civil Engineering, TU Delft, 2628 CN Delft, The Netherlands; 3Building Materials and Structures, TNO, 2629 JD Delft, The Netherlands; sayeda.nahar@tno.nl

**Keywords:** REOB, VTAE, waste oil, bitumen, oxidation, ageing susceptibility, molecular stability, low-temperature performance, rheology, lubricant additives, brittleness

## Abstract

The evolving bitumen market is increasingly complex due to variations in crude sources and transitions in refining processes, affecting the properties of bitumen. Unexpected additions of materials to alter bitumen’s properties could occur, where traditional PEN grade testing fails to detect modifications by inclusion of, for example, Re-refined Engine Oil Bottoms. This is the first study to comprehensively compare REOBs from European vs. North American sources and assess their effects on binder performance in a unified framework, performed by assessing the REOB-modified binders by identification, stability, compatibility, ageing susceptibility, and low-temperature properties. Two series of REOB-modified bitumen were prepared by blending 5, 10, and 15 wt.% REOB into hard grade bitumen. Results showed increased carbonyl formations (likely caused by lubricant additives) and phase instability during storage which can be attributed to saturates exudation. Rheological assessment demonstrated that REOB softens bitumen, although ageing causes a pronounced gain in stiffness. Low temperature rheological measurements showed that REOB-modified bitumen is prone to brittle fracture, suggesting a loss of relaxation properties. This study highlights that REOB is a material of inconsistent nature, with complex interactions with molecular groups of the base bitumen, causing increased ageing, phase instability, and brittle fracture susceptibilities.

## 1. Introduction

Climate change, geopolitical tensions, and changes in crude oil refinery equipment are constantly redefining the bitumen market, which now has evolved into a complex landscape in recent years. These changes impact bitumen composition and properties, especially in terms of workability, durability, and health and safety [1]. Such changes have a direct impact on mixture design, pavement durability, and recyclability at end of life. Inconsistencies arise when different grades of bitumen are produced by blending secondary streams of various origins, which often remain unspecified and unknown. One such blending component is Re-Refined Engine Oil Bottom (REOB) or vacuum tower asphalt extender (VTAE). The re-refining of waste engine oil produces REOB, a residuum increasingly available due to the global incentive to recycle engine oil [2,3]. With its lower viscosity compared to bitumen, REOB is used to produce softer grades. Alternatively, to REOB, many other oils have been applied over the years to effectively change bitumen’s viscosity, i.e., bio-oils like vegetable oils, tall oils, or other crude oil-derived oils like paraffinic or other aromatic oils. REOB is quite unique in this regard however, as it is seen as, much like bitumen, a waste residue from vacuum distillation. Due to its availability, this would make REOB an applicable modifier for bitumen. REOB is not used in the industry as compared to the quantities that are produced with refining waste engine oil. Burning or storage of this oil is unwanted, and the need to find proper use of this oil is wished for. With recent years increasing the awareness with regards to limiting environmental impacts, preventing the production of waste and increasing sustainability of the recycling of waste engine oil by using REOB in bitumen is deemed as a valid option. Nevertheless, research has shown inconclusive outcomes and raised concerns for the performance of REOB-modified bitumen, largely due to the variability in properties resulting from different feedstocks and dosages. REOB-modified bitumen has been reported to be more susceptible to ageing and low-temperature cracking [2,4,5,6,7].

In Europe, REOB does not have a unique identification number like a Chemical Abstracts Service (CAS) number and is sold as bitumen, unlike in the US, where it is traceable through its own CAS number [8]. This limit in traceability poses a challenge for asphalt contractors who may be unaware of the presence of REOB in the bitumen. Recent articles have reported poorly workable mixtures during asphalt production and premature damage in road surfaces in the Netherlands, where REOB was identified in the asphalt binder [8,9]. Additionally, one should be aware that REOB does not have a consistent composition, due to its nature. It is produced with the recycling of waste engine oil, a material that itself has no distinctive single composition (it can have slightly different viscosities and lubricant additives added to it). The standard PEN grade testing, Ring and Ball softening test, and PG system all do not detect the presence of REOB in binders, and the industry is currently unaware of its use. This emphasizes the need for awareness and knowledge of the performance of such binders within the European asphalt industry.

To evaluate the effect of REOB dosage and aiming to create a 70/100 grade binder out of harder grade binders, 10/20 grade binders were used as base bitumen for modification. Two series of REOB-modified bitumen were prepared in the laboratory using two hard grades of base bitumen from different sources, blended with REOB from a single source. To account for variability in REOB sources, REOB produced in the US is compared to two different batches produced in Italy.

The research objectives that this study tries to answer are formulated as follows:How can REOB be characterised?How does REOB alter hard PEN grade bitumen its chemical and rheological properties?What are the limitations of REOB modification?

The research approach applied in this study includes three physicochemical aspects as shown in Figure 1: identification or fingerprinting of REOB and its compatibility, the ageing susceptibility, and its impact on low-temperature behaviour. Answering these questions with this approach will lead to effective characterisation of REOB and present the drawbacks of its use, for both researchers and industry, as the attractiveness of the use of this residue oil is deemed to increase in the coming years.

## 2. Materials and Methods

### 2.1. Materials and Conditioning

In this study, two different series of (modified) bitumen were evaluated to compare the impact of REOB modification. Figure 2 presents an overview of the materials studied. Bit-K and Bit-P are both hard-grade bitumen with PEN grades of 20/30 and 10/20, respectively. These base bitumen are blended with REOB to create modified binders, which are then compared to two reference PEN 70/100 bitumen, Bit-J and Bit-O, each sourced from the same suppliers as Bit-K and Bit-P, respectively. Both binder bit-J and bit-K were produced using a Solvent De-Asphalting unit (SDA); this is especially visible in the very low asphaltene (and saturate) fractions in the SARA test performed (visible later in Section 3.1.3).

Binders bit-O and bit-P are known to be straight-run binders produced from a Venezuelan crude oil, with no additional refinery steps used, compared to bit-J and bit-K; this would explain the significantly higher levels of saturates and asphaltenes. In addition to the bitumen, three different REOBs, REOB-X, REOB-Y, and REOB-Z, were also evaluated, focusing on their chemical properties. The first two REOBs both originate from the same refinery in Italy, processed from different feedstocks, while REOB-Z is processed in the USA, thus again from another source.

Among the three REOBs, REOB-X was used to study the effect of REOB on bitumen properties. For the blend preparation, the base bitumen Bit-K and Bit-P were pre-heated to 163 °C, and REOB-X was preheated to 130 °C. Bitumen and REOB were blended using a low shear mixing assembly at a temperature of 130 °C for 15–20 min. This blending was performed at three different dosage levels: 5, 10, and 15 wt.%. As mentioned before, these bitumen have different PEN grades which are presented in Table 1. These PEN grades show that a dosage of roughly 15–20% is enough to reach the commercially desired 70/100 grade binder, in the case of using these two base-bitumen. However, the viscoelastic behaviour described at the end of Section 3.1.3, shows that a dosage of 10–15% is enough to reach similar behaviour as the 70/100 reference bitumen. This signifies a mismatch between the PEN grade indication for viscoelastic properties in the case of REOB-modified bitumen.

To assess ageing susceptibility, the binders were conditioned for short-term ageing followed by a long-term ageing protocol. For short-term ageing, the Thin-Film Oven Ageing method was applied, using PAV plates with a 50 g sample per plate, conditioned in an oven at 163 °C for 5 h. This was followed by placing the PAV plates in the Pressure Ageing Vessel (PAV) following the long-term ageing protocol at 100 °C and 20 bar pressure for 20 h. All binders were aged with the exact same time in the same oven, meaning differences between binder source and dosages can be properly evaluated, although only a single PAV cycle was applied.

### 2.2. Methods of Characterisation

For the evaluation of the chemical composition and the rheological behaviour of unmodified and REOB-modified bitumen, a combination of chemical, rheological, and mechanical characterisation tools have been used. These range from Attenuated Total Reflectance-Fourier Transform Infrared spectroscopy (ATR-FTIR) to Fractionation of Saturates/Aromatics/Resins/Asphaltenes (SARA), to Gel Permeation Chromatography (GPC), Fluorescence Microscopy (FM), Dynamic Shear Rheometer (DSR), Bending Beam Rheometer (BBR), and Dynamic Mechanical Analyser (DMA). In the following sections, the applied methodologies have been explained:


**ATR-FTIR spectroscopy**


A Perkin Elmer Frontier ATR-FTIR spectrometer (PerkinElmer, Shelton, CT, USA) was used to determine the characteristic chemical functional groups of the binders. One can obtain transmittance spectra in the mid-infrared region. This allows for the identification of different functional groups that are present in the binder once placed on the ATR crystal (in this case, a diamond crystal was used). In the region of 4000 cm^−1^ to 680 cm^−1^ wave number, twenty-four scans per spectrum, three measurements were performed per (modified) bitumen. However, when a binder deviated too much, it was removed from the group evaluated. For unaged binders, the sample was applied on the crystal with a slight pressure—to ensure good sample contact with the crystal. However, for the aged binders, a pressure gauge attached to the set-up was used to apply pressure to ensure good contact based on work from Mirwald et al. [10]. To obtain indices, the spectra were ATR modified, converted to absorbance spectra, (flat) baseline corrected, and normalised (between minimum and maximum). A full baseline area calculation was then applied to quantify all functional groups present in the binders, with distinct indices showing the specific peak areas as a fraction of the total peak area based on Hofko et al. [11]. The Index of Carbonyl Oxidation (ICO) was calculated by dividing the area of the carbonyl area by all the areas that were not connected to oxidation, i.e., aliphatic peak (stretching, bending, rocking), aromatic (stretching and bending vibrations). The Index of Sulfoxide Oxidation (ISO) was calculated in a similar manner.


**SARA Fractionation—Iatroscan**


The ASTM D4124-09(2018) [12] standard test method separates neat bitumen, REOB, and REOB-modified bitumen into saturates, aromatics, resins, and asphaltenes (SARA). The procedure involves sample spotting, separation of the fractions using increasing polarity of the solvents, and quantification using the Iatroscan MK-6s instrument (SES GmbH, Bechenheim, Germany). Samples are dissolved in toluene (10 mg/mL), and 1 µL is spotted on quartz chromarods (the stationary phase) using a semi-automatic spotter. Fractionation occurs in three stages with solvents of increasing polarity: n-heptane separates saturates, a 20:80 mixture of n-heptane and toluene separates aromatics, and a 5:95 mixture of methanol and dichloromethane separates resins. Asphaltenes remain at the application spot. Quantification is achieved through Thin Layer Chromatography-Flame Ionization Detection (TLC-FID), where chromarods are scanned with a hydrogen/air flame at 30s speed. The resulting ion counts produce a chromatogram, identifying fractions by characteristic peaks and determining their composition by peak areas. Each analysis is performed in triplicate for accuracy.


**Gel Permeation Chromatography (GPC)**


Waters AQCUITY GPC (Waters Corporation, Milford, MA, USA) was used to analyse the molecular mass distribution of bitumen, REOB, and REOB-modified bitumen in this study. The system consists of an autosampler, two detectors, a pump, and columns. By dissolving the binder sample in a solvent (i.e., tetrahydrofuran) and injecting it into a flowing stream, molecules are separated based on their sizes. Detection is performed using UV and refractive index detectors. The retention time in GPC indicates molecular size, which correlates with molecular weight. Polystyrene calibration standards were used to determine the molecular weight distribution. Due to limited budget, only single samples were measured. Variations between samples have therefore not been able to be determined.


**Fluorescence Microscopy (FM)**


A Nikon Eclipse Ci (Nikon Europe, Amstelveen, The Netherlands) brightfield, darkfield, and fluorescence microscope was used to obtain specifically fluorescence micrographs of smooth/cooled bitumen surfaces. Fluorescence mode used an excitation filter at 403/95 nm (353–452 nm), an emission filter (long pass) at 500 nm, and a dichroic mirror at 495 nm. As the fluorescence of the bitumen is considerably less than the reflection in the brightfield mode, an exposure time of 900 ms was used. Following the approach of Mirwald et al., a detailed description of the method can be found in [10].


**Dynamic Shear Rheometer (DSR)**


To characterise the rheological behaviour of the binders, the Modular Compact Rheometer MCR 502 DSR (Anton Paar, Graz, Austria) was used. The binder sample was placed between two parallel plates, with the bottom plate fixed and the top plate oscillating to apply shear strain. The frequency sweeps from 0.01 to 400 rad/s were conducted at temperatures ranging from −10 °C to 60 °C, in 10 °C intervals. Strain sweeps were conducted first to determine the linear viscoelastic region at each temperature level. For this study, an 8 mm diameter parallel plate was used from −10 °C to 30 °C, and a 25 mm diameter parallel plate was used from 30 °C to 60 °C. Master curves were constructed using the time–temperature superposition principle, with a reference temperature of 20 °C. Only single samples were measured, although measurements were taken for each temperature range. The consistency and overlap between changing temperatures were checked to confirm that the state of measurements was proper—whenever this was not the case, a new sample was measured.


**Bending Beam Rheometer (BBR)**


A BBR test was performed with a TE-BBR SD (CANNON, Cranberry, PA, USA) according to the NEN-EN 14771 [13] and evaluates the low-temperature stiffness and cracking potential of the binders, measuring midpoint deflection under a constant load of ~980 mN at various temperatures. The flexural creep stiffness and creep rate (m-value) are determined by recording deflections at 8, 15, 30, 60, 120, and 240 s. Creep stiffness indicates resistance to a constant load, while the creep rate represents changes in stiffness over time. Two samples were tested, one at each temperature around the critical temperature to determine it.


**Dynamic Mechanical Analyser (DMA)**


For dynamic tests at lowered temperatures, a DMA 3200 was used (TA Instruments, New Castle, DE, USA). The DMA exerts tensile forces on small bitumen samples, producing stress–strain diagrams from standard tensile tests. A small bitumen sample is placed between two stone columns (diameter 8 mm of “bestone”, a specific type of gravel used in the Netherlands exclusively for porous asphalt) at a controlled thickness of ~0.2 mm. This specimen assembly was placed in DMA and pulled apart at different strain rates of 0.002, 0.004, and 0.006 mm/s at temperature 10 °C conditions, and the cumulative responses of adhesive and cohesive behaviour are observed as distinct viscoelastic elastic properties of the binders. A more detailed description of the procedure can be found in [14]. Three replicates were measured to ensure accuracy.

Many testing conditions or experiments are represented by a single measurement. As such, no formal statistical analyses (e.g., ANOVA or *t*-tests) were performed, and the results are presented as representative values to highlight relative trends between samples.

## 3. Results

The following sections present the results obtained from the experimental tests on the binders, based on the methodologies previously described. The results are presented on three main aspects: the effect of REOB on fingerprinting and compatibility, the impact of REOB on ageing, and the influence of REOB on the low-temperature behaviour of the binder.

### 3.1. Fingerprinting, Stability, and Compatibility of REOB

A combination of chemical characterisation tools is used to distinguish REOB from bitumen. ATR-FTIR identifies the characteristic functional group related to REOB in the binder. Additionally, GPC measures the molecular weight distribution of pure REOB, indicating whether it consists of characteristic molecular sizes or has a broad distribution of varying sizes. The SARA method measures different molecular fractions present in REOB and (modified) bitumen based on polarity differences.

Compatibility was then assessed using SARA fractions, molecular weight distributions, viscoelastic behaviour, and fluorescence microscopy micrographs, all of which can indicate if, at the chemical level, REOB blends well with bitumen.

#### 3.1.1. Composition of REOB from Different Sources

Although the use of REOB has been common since the early 1980s, the changes in its production, source, and dosages are not regulated [2,3,15]. REOB, derived from the re-refining of waste engine oil, has an inconsistent nature depending on the production and origin of the feedstock.

In Figure 3, intensity differences of several functional groups present in the three analysed REOB sources can be observed. Conventionally, REOB is like bitumen in its hydrocarbon nature, showing clear aliphatic structures with the presence of alkenes (CH_2_) and alkanes (CH_3_), and aromatic structures with (C=C). However, more distinctive functional groups are indicated by the higher intensity of carbonyls (C=O) peaks, higher double-bonded carbon (C=C), polyisobutylene (PIB), silicon oil (Si-O-Si), and phosphate bonds (P-O-C). Among the three REOB sources, the spectra confirm that they consist of similar molecules, with no additional signature functional groups beyond the noted ones.

The primary difference lies in the intensity of these groups, especially between REOB-Y and REOB-Z. Additionally, REOB-X shows an increase in all its functional groups after 20 h of PAV ageing, indicating that REOB is also susceptible to oxidative ageing like bitumen. The peaks of PIB, Si-O-Si have been used previously to identify the presence of REOB (in bitumen) and to quantify dosages [16,17]. PIB is a commonly used polymer to increase the viscosity of engine oil, while Si-O-Si points towards the presence of silicon oil, a compound used in dampers of brakes, and commonly mixed in waste engine oil when replaced. Additionally, the P-O-C peaks that were pointed out in Figure 3 are functional groups specific for phosphorous based anti-wear additives, like for example zinc dialkyldithiophosphate (ZDDP). As the waste engine oil source can vary heavily for produced REOB, the amounts and differences in lubricant additives can differ drastically. ATR-FTIR, however, allows for a quick estimate of their overall presence and can give an indication of bitumen modification with REOB or other engine oil derivatives.

Figure 4 presents the molecular weight distribution of the three different REOBs, including the 20hPAV-aged variant of REOB-X. From this, one can expect a different effect from each of these REOBs on bitumen when modifying its viscosity. The 20hPAV ageing of REOB-X shows significant changes to its original distribution. Like bitumen, ageing results in an increase in the high-molecular-weight shoulder (mainly related to asphaltenes) and a decrease in the low molecular weight (‘maltene’) peak, however, in a distinct pattern. However, for REOB, this peak also shifts to the left, indicating that the maltenes tend to attain a higher molecular weight. This could very well be related to increased levels of oxidation found in this study and presented in Section 3.2.1.

#### 3.1.2. Stability of REOB and REOB-Modified Bitumen

SARA analysis was first carried out to determine the molecular fractions of saturates, aromatics, resins, and asphaltenes in REOB. This section addresses a significant issue observed in the composition of REOB. Since REOB is derived from a similar source as bitumen—crude oil—its polarity-driven molecular fractions stay similar, aside from lubricant additives. The SARA analysis presented in Figure 5 indicates that REOB primarily consists of saturates and asphaltenes, with a relatively low aromatic fraction, which may impact the molecular equilibrium and phase behaviour of REOB-modified bitumen. The high asphaltene fraction is unlikely to be similar to the insoluble compounds in n-heptane, as those from bitumen. In bitumen, they consist mainly of polyaromatic compounds, big clusters that have formed under higher pressures and with time in the crude oil. In engine oil and thus in REOB, such compounds were not present from the start and likely have not formed in the short time of their use. This fraction instead mainly consists of the insoluble fraction of polymers, soot, metal additives, and unburnt fuel.

Some of these REOB samples were analysed after 2 years, while the samples were stored at 5 °C and in the dark. Re-measurement revealed a significant change in the SARA fraction. This variation suggests that part of the low molecular weight molecules may have evaporated; there were changes in polarity possibly caused by reactive species or any irreversible changes that cannot be recovered through re-mixing. These findings raise a concern about the stability of REOB and, hence, the need for careful monitoring in relation to phase behaviour during long storage periods. It remains unclear whether this issue also arises when REOB is used to modify bitumen, as well as the potential differences in bitumen properties arising from various storage conditions.

#### 3.1.3. Compatibility of REOB with Bitumen

The observed instability in REOB suggests a closer study of the compatibility of REOB with bitumen is necessary. To accommodate for this, SARA fractions, GPC weight distributions, and DSR viscoelastic behaviour were used to evaluate the modified binders.

To start off, the four different SARA fractions shown in Figure 6 illustrating that the harder PEN grade bitumen, Bit-P and Bit-K, have relatively low saturate content. Bit-P has higher resin and asphaltene content than Bit-K. After the addition of REOB-X, there is a slight increase in saturates and aromatics, while resins and asphaltenes either decrease or remain similar to the base bitumen, and like the ageing of bitumen, the REOB-modified bitumen shows a loss of aromatics, an increase in resins, and an increase in asphaltenes. However, unlike bitumen, there is a noticeable decrease in saturates (which usually remain constant for unmodified bitumen); next, the loss in aromatics is much more substantial than the decreases measured for unmodified bitumen.

This raises the question of whether REOB is undergoing rapid changes and hence affecting the fractions or if REOB accelerates the ageing process in bitumen. The change in saturates appears distinct from typical ageing effects which may be related to the exudation of saturates from REOB-modified bitumen, as observed in previous research [18]. It is thus possible that the types of saturates introduced by REOB are different in nature and less compatible with the bitumen.

The addition of REOB not only changes the polarity of different molecules in bitumen but also affects the molecular weight distribution. Figure 7 shows the molecular weight distributions of unaged blends on the left (a and c) and aged versions on the right (b and d).

The distributions reveal that adding REOB at 10% dosage increases molecular weight, with the ‘maltene’ peak shifting towards higher molecular weight. This shift suggests an initial ageing effect. On its own, REOB does not have such a high average molecular weight. When blended with bitumen, however, two things may occur: the lighter fractions of REOB may evaporate, or REOB may interact with bitumen to form larger molecular clusters. Either way, the average molecular weight of the blend increases. Still, an increase in molecular weight alone does not directly determine binder viscosity; molecular interactions remain the key factor. However, an increase in molecular weight alone does not directly affect the viscosity of the binder; the interactions between molecules play a crucial role. The 20hPAV aged distributions of the bitumen and blends show that the molecular weight distribution of REOB-modified bitumen overlapped with those of the aged base bitumen.

This observation suggests that the molecular alteration introduced by REOB at a 10% dosage (with these specific bitumen and REOB) is lost after 20hPAV ageing. From FTIR and FM results in the research, it is known that REOB remains in the binder. This implies that the addition of REOB creates an initial imbalance in the modified bitumen, which is quickly “recovered” after ageing, as reflected in the molecular weight distributions. This could explain why some researchers have encountered difficulties—using REOB to blend with RAP and RAS, particularly when analysing GPC molecular weight distributions [19].

Frequency sweep analysis with DSR, produced master curves of the complex shear modulus and phase shift angle for unmodified and REOB-modified bitumen, as shown in Figure 8. The left side (a and c) represents the unaged condition, while the right side (b and d) represents the 20hPAV-aged condition. The master curves were obtained by CAM-model fitting for both properties by only applying horizontal shifting, and no vertical shifting was applied to the data.

From the viscoelastic testing, it can be found that REOB can effectively lower the stiffness of bitumen while also increasing its phase shift angle; this is shown by the shifts the curves undergo on the left side of Figure 8. Here one can see what was stated before in Section 2.1.; the bit-P+15% comes close to the rheological behaviour of bit-O at 70/100, while bit-K+10% is sufficiently close to the behaviour of bit-J. One can also see that REOB does not affect the rheological characteristic behaviour of bitumen; the shapes of both the stiffness and the phase angle are relatively the same, and their time–temperature relation was merely affected after increasing REOB dosage. This is likely the reason why several papers have shown good results for REOB-modified bitumen [20,21,22], as it indeed does what it is intended to do: soften the bitumen.

In the 20hPAV-aged condition, higher REOB dosages exhibit different effects compared to lower ones. The Bit-P+5% and Bit-P+15% curves show almost a complete overlap in both stiffness and phase angle, suggesting that the ageing process negates the softening effect of REOB regardless of the dosage level. Comparing this to the behaviour of the unmodified reference bitumen Bit-O and Bit-J, it is evident that the presence of REOB influences the ageing behaviour of the bitumen, suggesting an accelerated increase in stiffness and decrease in phase angle.

To show more clearly the viscoelastic behaviour of these binders at higher temperatures (a quick indication of the impact of REOB on the binders’ consistency), complex dynamic viscosity is plotted in Figure 9. These are calculated from $\eta^{*}=G^{*}/\omega$, showing that for binder series 1, around 15% of REOB was needed to lower the viscosity to as low as the reference binder, and around 10% for series 2. After ageing, specifically bit-O shows to have much better resistance. At these higher temperatures, bit-J undergoes similar changes as the REOB-modified binder bit-K+10%.

#### 3.1.4. Micrographs of REOB-Modified Bitumen

To assess the dispersion of the REOB phase and phase behaviour of REOB-modified bitumen, fluorescence microscopy is applied. If REOB is different in fluorescence, it would be visible as a separate phase in bitumen from the micrographs. Micrographs of the two REOB-modified blends with a higher dosage, Bit-P+15% are presented in (c) and (d) of Figure 10, while (a) and (b) show the micrograph of Bit-K+15%. Unmodified bitumen, although not shown here, owns a fully homogeneous fluorescent field, and therefore one can assume that this dispersed phase was brought by the REOB directly.

These micrographs reveal a dispersed circular phase in the REOB-modified bitumen, shown by distinct fluorescence, suggesting the presence of a separate phase throughout the bitumen matrix. This corresponds closely to previously observed dispersed phases in the REOB-modified bitumen surface topography obtained with Atomic Force Microscopy (AFM) [23,24]. Previous studies raised concern for possible incompatibility of REOB in bitumen in such blends, as the distinct phase observed tended to change further upon ageing.

### 3.2. Ageing Susceptibility of REOB-Modified Bitumen

To evaluate the ageing of bitumen, two main aspects were considered: oxidation and change in rheological behaviour. FTIR was used to measure different levels of carbonyls and sulfoxides, indicating degrees of oxidation, while DSR allowed for plotting unmodified Blackspace diagrams to show the effect of ageing on viscoelastic properties.

#### 3.2.1. Oxidation Represented by Carbonyl and Sulfoxide Formation

ATR-FTIR was used to obtain mid-infrared spectra, which were then modified to obtain indices (see Section 2.2. for the procedure) indicating the presence of carbonyl, sulfoxide, aromatic, or other functional groups. These indices can show changes in intensities when comparing different binders. The oxidation of bitumen with ageing is primarily attributed to the formation of carbonyls (C=O) and sulfoxides (S=O), as shown in Figure 11.

The ageing of pure REOB shows an increase in carbonyl and sulfoxide contents after 20hPAV ageing, but not exceeding bitumen. However, a trend is observed in the 20hPAV-aged REOB-modified bitumen: the higher the dosage of REOB, the higher the rise in the carbonyl index. For series 1, this trend is observed initially, after blending with REOB. For series 2 this happens only after 20hPAV ageing. In contrast, the rise in sulfoxides is not related to REOB dosage, in the case of long-term ageing. This suggests that REOB influences oxidation mechanisms in bitumen, specifically increasing the formation of carbonyls. Such increased oxidation rates have been reported in earlier studies [6,25]. It is unlikely that REOB undergoes severe ageing, as it primarily consists of saturated naphthenic and iso-paraffinic compounds rather than waxes or long-chain paraffins. Instead, it likely influences the carbonyl formation mechanism in bitumen ageing. Given its high polymer and antioxidant content, REOB can significantly impact the reaction mechanisms involved in bitumen ageing.

#### 3.2.2. Rheological Behaviour of REOB-Modified Bitumen Changing with Ageing

The viscoelastic behaviour of (modified) bitumen also reflects the effect of ageing. Using DSR frequency sweep data, Blackspace diagrams were plotted and presented in Figure 12, demonstrating the effect of ageing by the lowering of the curve and shifting towards the left.

As seen with the Master Curves in Figure 8, the initial modification of base bitumen with REOB does not alter the characteristic viscoelastic behaviour: the Blackspace diagrams keep the same shape, and a direct overlap is observed. Initially, REOB impacts only the time–temperature relation of the viscoelastic properties of the bitumen. However, looking at the 20hPAV aged bitumen on the right side of Figure 12, in (b) and (d), distinct changes are observed in the curves of REOB-modified bitumen. Notably, the series 1 Bit-P modified bitumen shows that the higher REOB dosage leads to a higher loss in phase shift angle. Although the bit-K blends seem to overlap slightly better, the bit-K+15% (purple symbols) are slightly lower than the others. This indicates that while REOB initially softens the bitumen, it accelerates the ageing process, resulting in more pronounced changes in viscoelastic properties.

### 3.3. Low-Temperature Behaviour of REOB-Modified Bitumen

The low-temperature behaviour of REOB-modified bitumen is evaluated using DSR, BBR, and DMA. The Cole–Cole diagram is used to show frequency sweep data from the DSR in the low-temperature region; the critical temperatures are determined using BBR, and the stress–strain tensile behaviour of thin bitumen films is evaluated with DMA at low temperatures.

#### 3.3.1. Viscoelastic Low-Temperature Region in Cole–Cole Diagrams

The DSR data are further analysed using Cole–Cole diagrams, which can effectively plot frequency sweep data in the low-temperature region. These diagrams show how the loss modulus (viscous component, G″) reaches a plateau while the storage modulus (elastic component, G′) continues to increase [26,27].

These Cole–Cole diagrams, shown in Figure 13, demonstrate that initial REOB modification alters the maximum attainable storage modulus for both binders. However, after 20hPAV ageing, these effects become even more pronounced. This suggests the low-temperature properties of REOB-modified bitumen show a loss of viscous property, causing the binders to show a different response than expected. Additionally, it is observed that higher or lower REOB dosage has little to no effect, as the curves in the Cole–Cole diagrams overlap.

#### 3.3.2. Low-Temperature Stiffness and Relaxation Properties

The BBR data are evaluated to determine the critical temperatures for the tested (modified) bitumen beams. These critical temperatures include Critical Stiffness (T_c_(S)), Critical Relaxation (T_c_(m)), and Critical Difference (ΔT_c_) temperatures. The lower these values, the lower the temperature of the material must be to reach either a stiffness of 300 MPa or a slope in the stress/strain curve of 0.3 after 60 s of applying a load of 1 N on the sample.

Table 2 presents the mentioned critical temperatures for both unmodified and REOB-modified bitumen, unaged, and 20hPAV aged conditions. It is evident that by increasing the dosage of REOB, one can effectively lower both the stiffness and relaxation critical temperatures. Notably, in series 2 Bit-K modified bitumen, the ΔT_c_ tends to decline slightly even without ageing. This trend is also observed after 20hPAV ageing, with higher REOB dosages resulting in more negative ΔTc values.

This indicates that with ageing, REOB-modified binders gain in stiffness but lose significantly in relaxation capabilities, as shown by the shifts in the presented critical temperatures. This suggests susceptibility to both ageing and the potential low-temperature cracking, which has been previously identified as a problem with REOB-modified binders, according to other research [4,5,6,25,28,29].

#### 3.3.3. Adhesion/Cohesion Behaviour at Low Temperatures

The adhesion and cohesion behaviour of thin bitumen films was analysed using DMA. Tensile pull tests were conducted at 10 °C with three different strain rates: 0.002, 0.004, and 0.006 mm/s. The resulting stress/strain behaviour is presented in Figure 14.

The stress/strain diagrams reveal a distinct difference between REOB-modified and unmodified bitumen. Both reference bitumen Bit-J and Bit-O exhibit significantly lower stress values, indicating a softer and more viscous behaviour at the 10 °C testing condition. In contrast, the REOB-modified bitumen demonstrates different behaviour, as Bit-K+10% fails to develop a complete stress/strain curve and breaks in a brittle manner, whereas Bit-P+10% shows slightly softer, though not significantly so. This observation contrasts with the overall rheological behaviour presented in Figure 8 that had shown that bit-K+10% had a perfect overlap with bit-J and bit-O was not far off from bit-P+10%. This begs the question: does REOB cause embrittlement of bitumen? This seems a valid concern, as the susceptibility of REOB-modified bitumen to turn from sol to gel-phase bitumen and a risk of physical hardening effects were noticed before [5,28]. This seems reasonable on its own; however, this was not seen in DSR measurements and should have been repeated in both cases. The main thought currently is the interaction of REOB-modified bitumen to the presence of mineral filler. Although REOB seems to disperse properly in bitumen itself, some separate phase is still visible as seen in 3.1.4. Both porosity of the filler and acidity of the binder could be properties directly related to the stiffening effect observed for these mastics. Further analysis is needed of binders with different acidity levels (for example by using Total Acid Number (TAN) test) and varying porosity of filler material. In mixture design, one should account for such a stiffening to occur in early service life.

## 4. Discussion

With the results analysed, this section specifically discusses what was presented before, summarising the observations and discussing the results more generally.

### 4.1. Interpretation of Fingerprinting, Stability, and Compatibility Results

REOB identification in bitumen was achieved through the presence of characteristic functional groups such as PIB, Si-O-Si, and P-O-C in FTIR spectra. The presence of these functional groups points towards lubrication additives. Which originate from the engine oil and one can notice that little variation exists in the types of functional groups across these three REOB sources; however, the intensity of these groups varies significantly. GPC has shown there is a high variance in the molecular weight distributions of the different REOB sources, indicating that the ‘maltene’ peak of REOB lies at a slightly higher average molecular weight than of bitumen. Additionally, the ageing of REOB increases oxidation rates, changes the polarity of molecules, and shifts the mentioned ‘maltene’ peak.

Regarding molecular stability as assessed by SARA analysis, it was observed that REOB undergoes significant changes in its molecular fractions while stored at cold (5 °C) and dark conditions, like bitumen storage conditions. However, the drastic changes in molecular fractions seen in REOB under the same conditions suggest a lack of molecular stability compared to bitumen. This instability could be due to component evaporation, irreversible changes in fractions, or an accelerated ageing mechanism: any one or combination of which may contribute to the shifts in SARA fractions.

The compatibility assessment of REOB to modify bitumen has shown that there is a significant influx of saturates to the bitumen. During the ageing of the blends, there is a higher loss of aromatics and a higher gain in resins and asphaltenes. However, the saturate fraction appears to decrease, which may relate to an earlier observation of REOB-modified bitumen, where saturates tend to exudate from the binder [18].

To evaluate which compounds specifically exudate from the REOB, but also from the modified bitumen, Gas Chromatography Mass Spectrometry could be performed. Much could be elucidated; future studies should include such evaluation techniques to improve the understanding of the stability of this oil and modified binders.

### 4.2. Implications of Ageing Susceptibility in REOB-Modified Bitumen

The ageing susceptibility of REOB-modified bitumen appears to increase as oxidation rates rise with higher REOB dosages, shown by the formation of carbonyls, but not for sulfoxides. It is unlikely that REOB undergoes severe ageing itself, as it primarily consists of saturated naphthenic and iso-paraffinic compounds rather than waxes or long-chain paraffins. Instead, it likely influences the carbonyl formation mechanism in bitumen ageing. Given its high polymer and antioxidant content, REOB can significantly impact the reaction mechanisms involved in bitumen ageing. This makes the choice of the base bitumen critical when modifying with REOB. Furthermore, the viscoelastic behaviour of REOB-modified bitumen initially demonstrated an overall effective softening, directly impacting the time–temperature viscoelastic response. Upon ageing, this softening effect diminishes, and the higher the dosage, the larger the loss is upon ageing, showing the influence of REOB on increasing the ageing susceptibility of a bituminous binder.

### 4.3. Evaluation of Low-Temperature Performance of REOB-Modified Bitumen

Lastly, the low-temperature behaviour of REOB-modified bitumen was characterized by DSR, BBR, and DMA. The Cole–Cole diagrams showed a lower maximum loss modulus at similar storage modulus levels for REOB-modified bitumen. Additionally, REOB-modified bitumen exhibits increased susceptibility to changes in critical temperatures with ageing, where higher dosages lead to more significant changes in ΔTc. However, the DMA testing reveals that REOB-modified bitumen tends to show more brittle-like behaviour and allows for higher stress build-up compared to reference bitumen. This suggests that REOB modification alters the base bitumen in such a way that it loses relaxation properties at low temperatures, an effect not directly noticeable with BBR. Future work could confirm this observation by evaluating REOB-modified bitumen using 4 mm DSR testing and conducting fatigue tests with DMA.

### 4.4. Study Limitations and Recommendations

This study evaluated only a limited number of REOB sources and base binders. Specifically, two Italian and one US-produced REOB were compared, which differed in lubricant additive concentrations as identified by FTIR. Two hard PEN grade bitumen were modified with these REOBs to illustrate the dependency of compatibility on the base binder. Given the varying nature of REOB, its compatibility and impact on ageing susceptibility may differ substantially between sources. Identifying which REOB properties determine phase stability, storage stability, low-temperature performance, and ageing resistance requires evaluation of a broader range of materials.

Additionally, many experiments were conducted with a single replicate due to material and time constraints, and only one PAV ageing cycle was applied; thus, results should be interpreted as indicative trends

Although phase separation and ageing susceptibility were observed, the underlying mechanisms remain unresolved. Future studies should assess the influence of lubricant additive types and concentrations on oxidation pathways and molecular interactions with base bitumen. Once the role of REOB source and base binder compatibility is better understood, mixture-level studies can be undertaken, followed by field validation to examine the influence of environmental conditions relative to laboratory findings.

These limitations also have implications for European regulations and standards. At present, specifications do not explicitly account for the variability of REOB composition and its potential influence on binder ageing and performance. Expanding the evidence base through larger datasets and field studies would support more informed regulatory guidance on the safe and sustainable use of REOB in asphalt binders.

## 5. Conclusions

From this study, the following conclusions can be drawn:REOB can be detected in bitumen through its distinctive functional groups, which are linked to residual lubricant additives. However, the intensity of these bands is small, and techniques such as NMR are more suitable for quantification.REOB significantly impacts the ageing susceptibility of bitumen, suggesting that it directly influences oxidation mechanisms of the base binder. Carbonyl growth increases substantially with higher REOB dosages, both after initial blending and after 20h PAV ageing, while sulfoxide growth remains unchanged. The exact mechanism is unclear, but antioxidants, trace metals, and residual polymers in REOB are believed to play a role.Storage stability of REOB, and therefore also that of REOB-modified bitumen, demands attention. REOB undergoes large polarity changes in its molecular composition (SARA fractions), showing a loss of saturates and a gain in asphaltenes. Micrographs confirm a dispersed phase across the surface. Unlike bitumen asphaltenes, these fractions consist mainly of soot, metal additives, and unburned fuel.Hard PEN-grade bitumen (15/20) can be modified with 10–15 wt.% REOB to reach a 70/100 grade binder with desirable viscoelastic properties. However, these benefits degrade more rapidly with ageing than in reference binders. The U.S. state DOT recommendation to limit REOB to 8 wt.% therefore seems reasonable, although such a restriction makes it difficult to soften hard PEN-grade binders to the 70/100 grade.DSR tests showed that REOB-modified samples had a significant decrease in loss modulus, while DMA revealed increased brittleness. These effects were not captured by BBR, suggesting that REOB used to decrease the low-PG grade may introduce unexpected complications.Because REOB is produced from variable waste engine oil sources, its properties, and those of REOB-modified bitumen, are unpredictable. Phase instability (likely from its naphthenic and paraffinic composition) and increased ageing susceptibility (due to lubricant additives such as antioxidants) raise concerns for use in hard PEN binders. Low-temperature tests also showed hardening, indicating that REOB does not effectively improve the low-temperature PG. REOB should therefore be identified in unknown binders, and these issues specifically addressed when considering REOB-modified bitumen.

## Figures and Tables

**Figure 1 materials-18-04825-f001:**
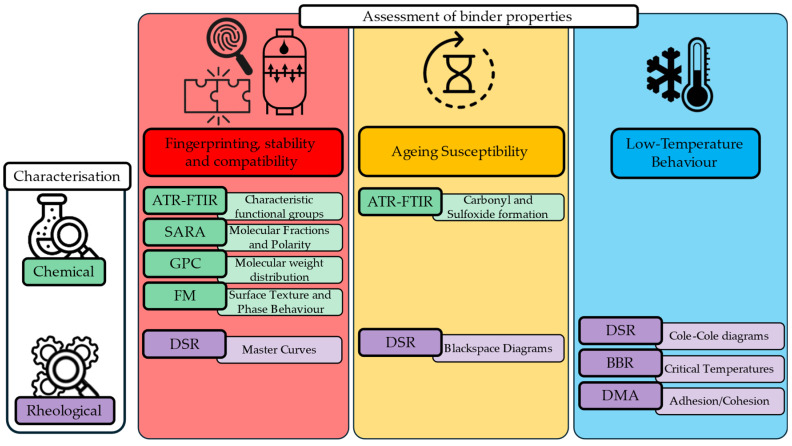
Research approach to evaluate REOB-modified binders.

**Figure 2 materials-18-04825-f002:**
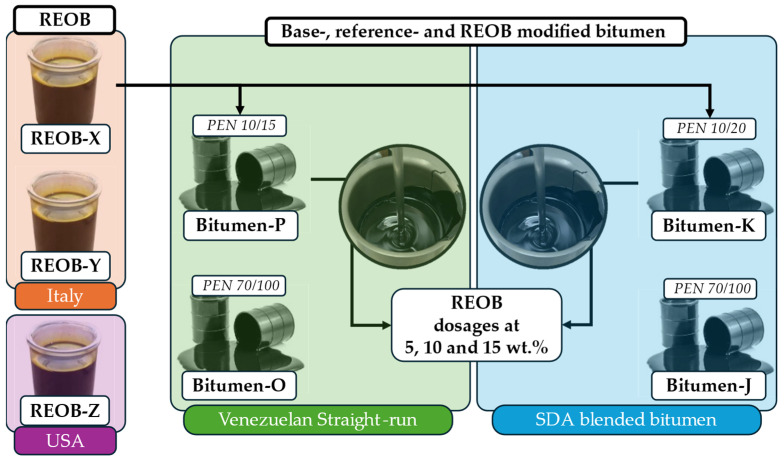
Overview of materials: REOB, bitumen, and REOB–bitumen blends.

**Figure 3 materials-18-04825-f003:**
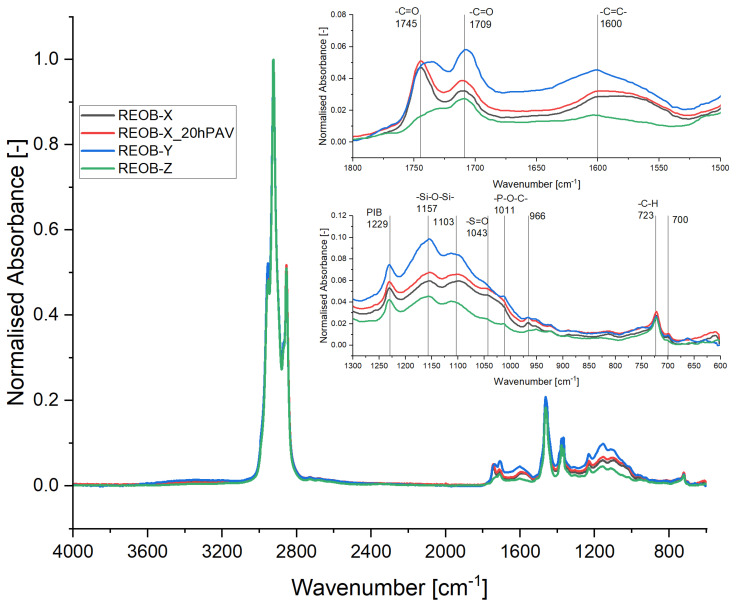
Fingerprinting REOB and assessment of ageing using ATR-FTIR spectra of unaged and one aged REOB after normalization.

**Figure 4 materials-18-04825-f004:**
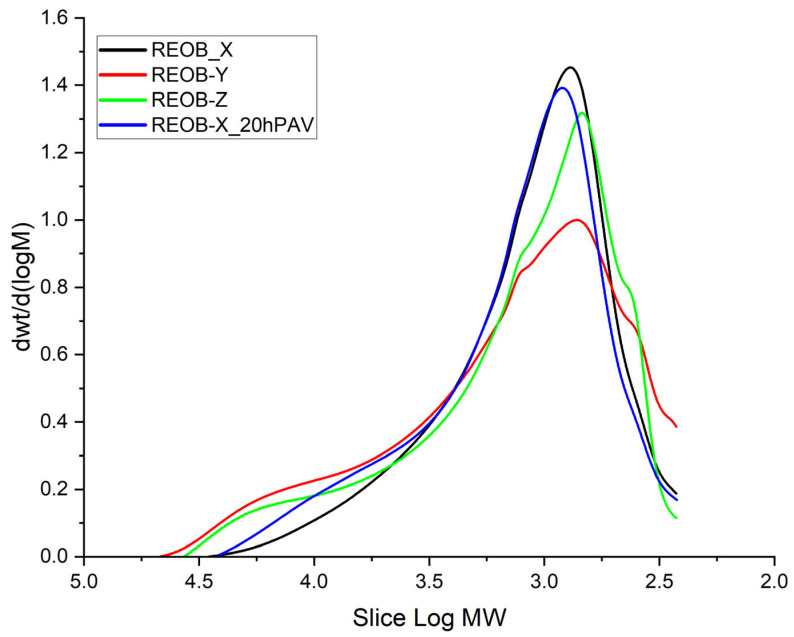
Apparent molecular weight distributions of three different REOB and one 20hPAV-aged REOB.

**Figure 5 materials-18-04825-f005:**
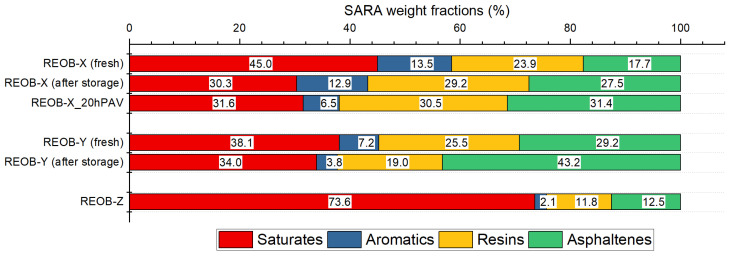
Overview of SARA fractions of REOB at different lifetime stages.

**Figure 6 materials-18-04825-f006:**
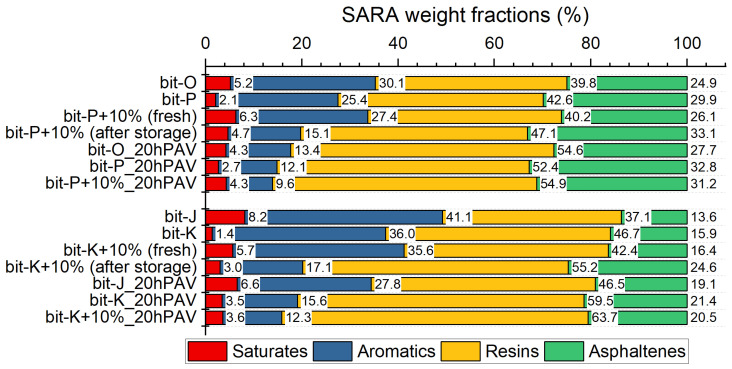
SARA fractions of base bitumen, REOB-modified bitumen, and reference bitumen.

**Figure 7 materials-18-04825-f007:**
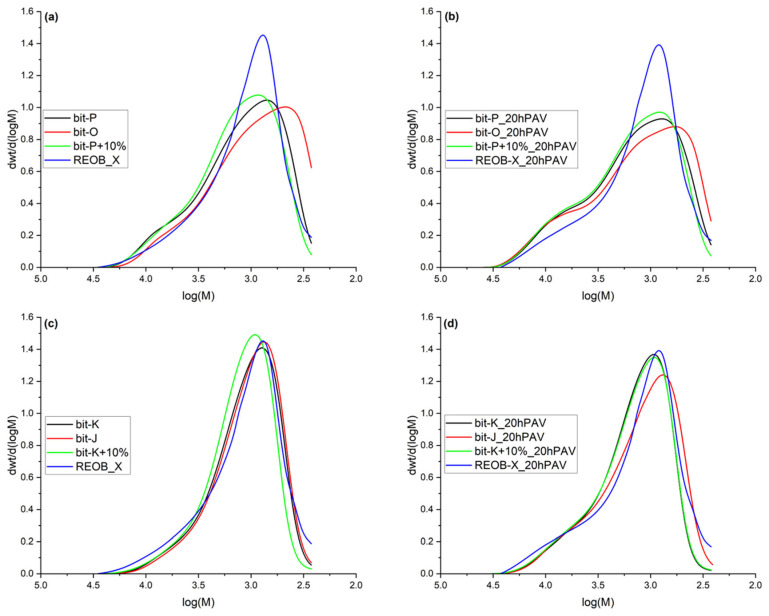
Molecular weight distributions of bitumen, REOB and REOB-modified bitumen; for series 1 (**a**) unaged, (**b**) 20hPAV aged and series 2 (**c**) unaged and (**d**) 20hPAV aged.

**Figure 8 materials-18-04825-f008:**
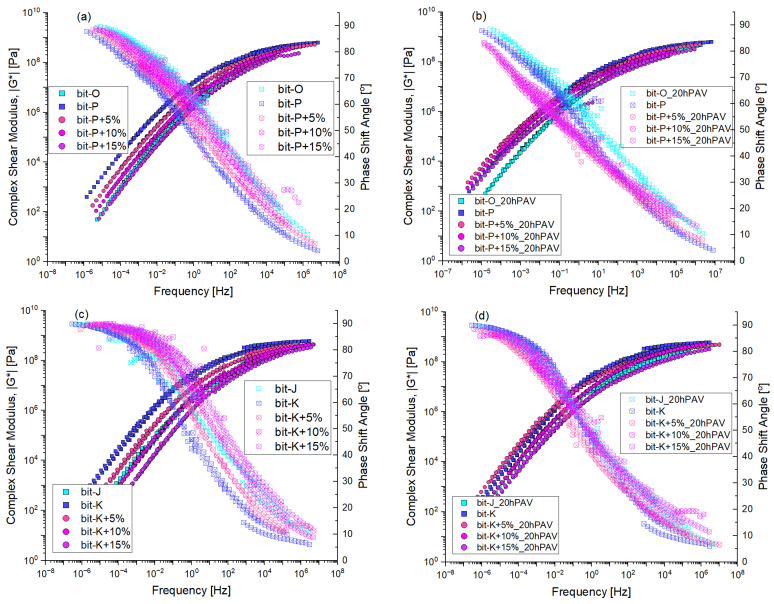
Frequency sweep data of complex shear modulus and phase shift angle; for series 1 (**a**) unaged, (**b**) 20hPAV aged and series 2 (**c**) unaged and (**d**) 20hPAV aged.

**Figure 9 materials-18-04825-f009:**
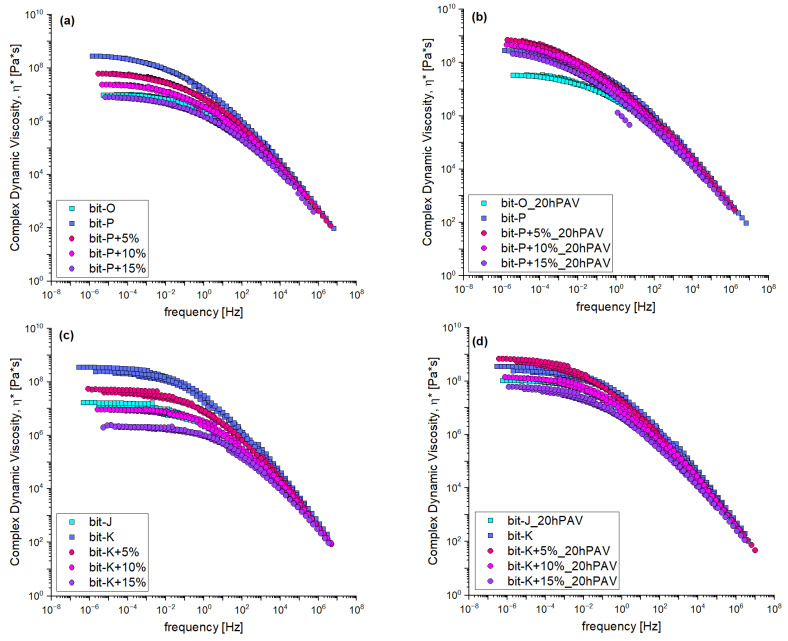
Complex dynamic viscosity from master curve fitted frequency sweep data, showing the high temperature viscosity; for series 1 (**a**) unaged, (**b**) 20hPAV aged and series 2 (**c**) unaged and (**d**) 20hPAV aged.

**Figure 10 materials-18-04825-f010:**
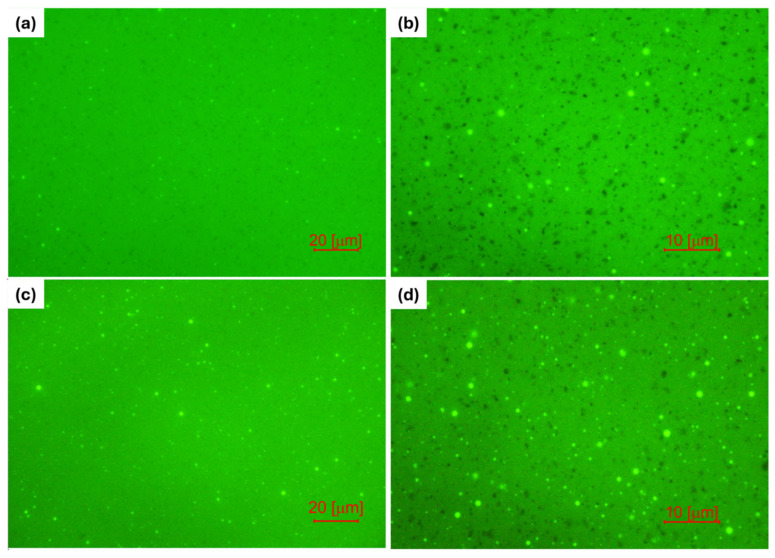
Fluorescence microscopy micrographs at (**a**) 40× and (**b**) 100× magnification of bit-K+15% and vice versa (**c**) 40× and (**d**) 100× magnification for bit-P+10%; showing a dispersed circular fluorescent phase throughout the bitumen.

**Figure 11 materials-18-04825-f011:**
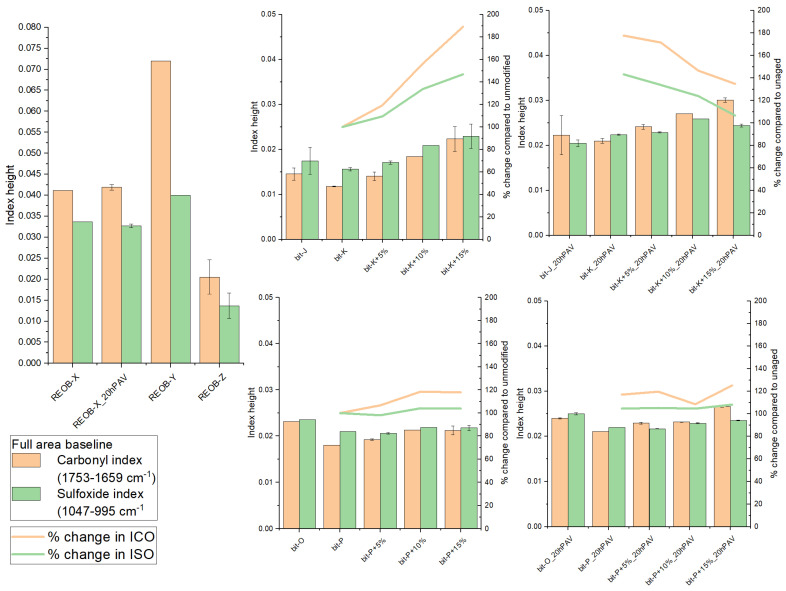
Indices for ATR-FTIR modified spectra REOB-modified bitumen.

**Figure 12 materials-18-04825-f012:**
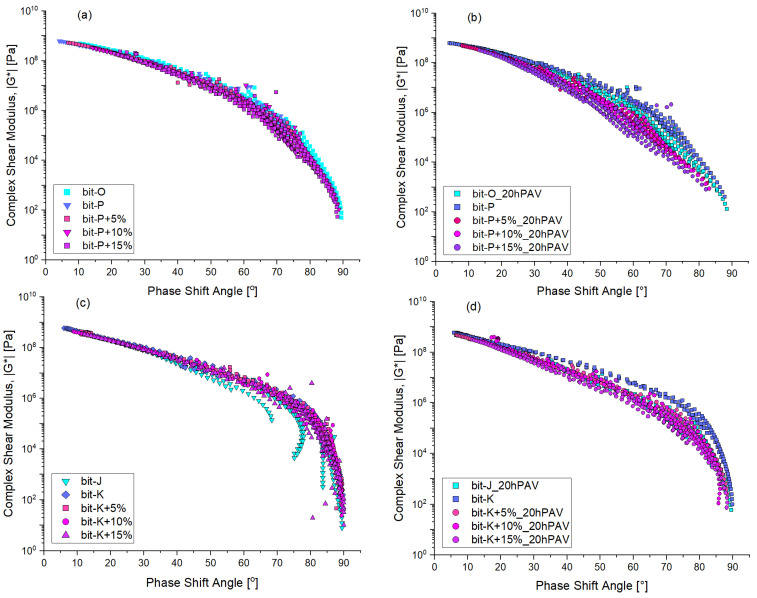
Rheological behaviour with ageing visualised in black space diagrams; for series 1 (**a**) unaged, (**b**) 20hPAV aged and series 2 (**c**) unaged and (**d**) 20hPAV aged.

**Figure 13 materials-18-04825-f013:**
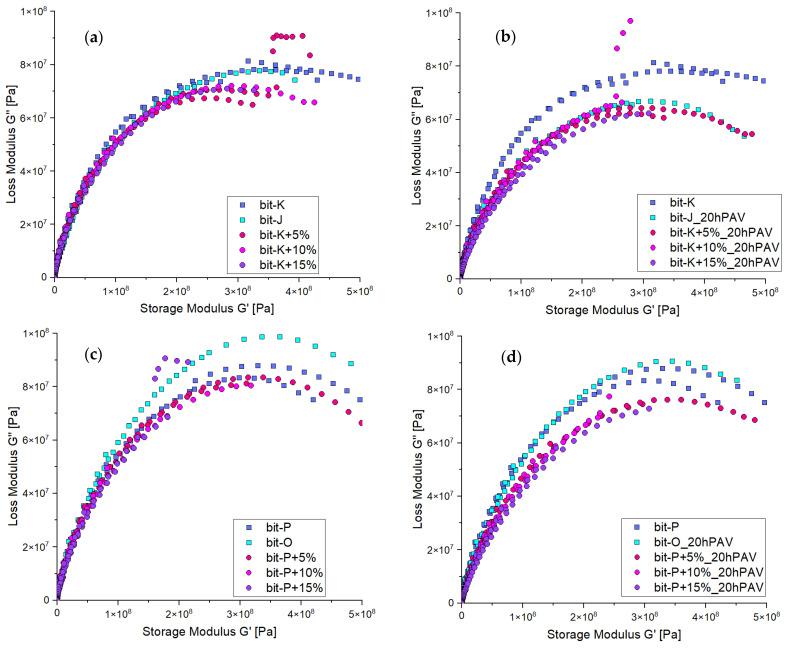
DSR results in low-temperature region visualised with Cole–Cole diagrams; for series 1 (**a**) unaged, (**b**) 20hPAV aged and series 2 (**c**) unaged and (**d**) 20hPAV aged.

**Figure 14 materials-18-04825-f014:**
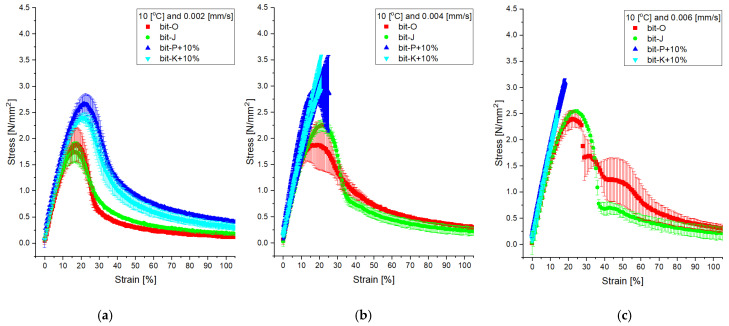
DMA test at 10 °C and (**a**) 0.002 mm/s strain rate for all four bitumen, (**b**) at 0.004 mm/s, and (**c**) the 0.006 mm/s.

**Table 1 materials-18-04825-t001:** Unmodified and REOB-modified bitumen their PEN grades and softening temperatures.

Samples Series 1	PEN at 25 °C, 0.1 mm	Softening Point, °C	Samples Series 2	PEN at 25 °C, 0.1 mm	Softening Point, °C
Bit-O	88	45.6	Bit-J	95	46.5
Bit-P	16	63.9	Bit-K	20	56.6
Bit-P+5%	25	58.9	Bit-K+5%	27	50.9
Bit-P+10%	39	55.3	Bit-K+10%	53	48.1
Bit-P+15%	61	50.5	Bit-K+15%	98	43.9

**Table 2 materials-18-04825-t002:** BBR measured T_c_(S), T_c_(m) and ΔT_c_ values for unaged and 20hPAV aged, unmodified, and REOB-modified bitumen.

Samples Series 1	T_c_(S), °C	T_c_(m), °C	ΔT_c_, °C	Samples Series 2	T_c_(S), °C	T_c_(m), °C	ΔT_c_, °C
Unaged							
Bit-O	−30.1	−33.6	3.5	Bit-J	−27.0	−29.2	2.2
Bit-P	−17.7	−20.4	2.7	Bit-K	−15.6	−18.2	2.5
Bit-P+5%	−23.1	−25.5	2.4	Bit-K+5%	−19.5	−21.8	2.3
Bit-P+10%	−27.5	−30.3	2.8	Bit-K+10%	−25.4	−27.3	1.9
Bit-P+15%	−32.4	−34.5	2.1	Bit-K+15%	−30.5	−31.9	1.3
20h PAV							
Bit-O	−27.2	−30.2	3.0	Bit-J	−25.3	−23	−2.2
Bit-P				Bit-K			
Bit-P+5%	−22	−21.2	−0.8	Bit-K+5%	−17.9	−16.7	−1.2
Bit-P+10%	−26.1	−24.8	−1.3	Bit-K+10%	−24.0	−22.6	−1.4
Bit-P+15%	−31	−28.8	−2.3	Bit-K+15%	−27.9	−25.3	−2.6

## Data Availability

The original contributions presented in this study are included in the article. Further inquiries can be directed to the corresponding authors.

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
