# Peer review of "Understanding the Limitations of Modifying Bitumen with Re-Refined Engine Oil Bottom (REOB)"

_materials, 2025, doi:10.3390/ma18214825_

Round 1
Reviewer 1 Report
Comments and Suggestions for Authors
General Assessment
The manuscript addresses a relevant and timely issue. However, improvements are needed in terms of clarity, novelty justification, and methodological rigor.
Major Comments
- Novelty and Significance: Please clarify the unique contribution of this study compared to existing literature. Highlight differences between European and US REOB sources as a novel aspect.
- Methodological Rigor: Number of replicates per test (DSR, BBR, DMA) is not clearly stated. Please specify and report error bars or standard deviations. Also, only one PAV ageing cycle was applied—this should be acknowledged as a limitation.
- Statistical Analysis: No formal statistical analysis is provided. Consider including ANOVA, t-tests, or confidence intervals, or justify why statistical tests are not feasible.
- Results and Interpretation: Some conclusions seem overstated. Please tone down or provide additional replicates.
- Discussion and Limitations: Add a section explicitly addressing study limitations (sample size, number of REOB sources, lack of field validation). Expand implications for European regulations and standards.
- Literature Coverage: Reduce reliance on self-citations. Include more independent studies from other regions (Asia, South America).
Minor Comments
- Correct all formatting errors, like references.
- Simplify long sentences in Methods and Discussion for better readability.
- Revise the Abstract to emphasize novelty more clearly.
- Add a note on environmental aspects of REOB use (sustainability, waste management).
- Make the Conclusions section more concise—avoid repetition of Discussion.
This manuscript requires minor language revision. Sentences that are too long need to be reviewed and shortened appropriately.
Author Response
Thank you for your extensive list of comments and recommendations. For our reply, please see attachment.

Reviewer 2 Report
Comments and Suggestions for Authors
The manuscript presents a study on REOB as a bitumen modifier. The subject is relevant, and significant findings are presented. Some points to be discussed:
- Give references to experimental methods, especially for SARA fractionation and FTIR.
- Chemical characterization, GC-MS, for instance, can elucidate changes in the composition of the mixture over storage time.
- Authors claim “The distributions reveal that adding REOB at 10% dosage increases molecular weight, with the ‘maltene’ peak shifting towards higher molecular weight.” The Changes in molecular weight are not wholly justified.
- The authors use a frequency sweep test to conclude on the ageing behavior of the bitumen. Please show the flow curve and viscosity curve for each sample. Also, time-dependentnt curve must be shown.
Author Response

(The authors gave the same response as above.)

Reviewer 3 Report
Comments and Suggestions for Authors
The work is undoubtedly relevant and performed at a high technical level.
Is there just one question?
The authors indicate that, according to IR spectroscopy data, REOB contains compounds with Si-O-Si and P-O-P bonds. The authors claim that compounds with Si-O-Si and P-O-P bonds are motor oil additives. I wonder what these substances are and how much of them are present in REOB. Or at least how much solid residue is present in REOB if it is burned (SiO2 and possibly P2O5 oxide or PO43- salts).
Author Response

(The authors gave the same response as above.)
